# Understanding Traffic Accidents among Young Drivers in Qatar

**DOI:** 10.3390/ijerph19010514

**Published:** 2022-01-04

**Authors:** Faris Tarlochan, Mohamed Izham Mohamed Ibrahim, Batool Gaben

**Affiliations:** 1Department of Mechanical and Industrial Engineering, Qatar University, Doha P.O. Box 2713, Qatar; bg1102639@student.qu.edu.qa; 2Department of Clinical Pharmacy and Practice, College of Pharmacy, QU Health, Qatar University, Doha P.O. Box 2713, Qatar; mohamedizham@qu.edu.qa

**Keywords:** risky driving behavior, traffic accidents, traffic violations, young drivers, traffic safety

## Abstract

Young drivers are generally associated with risky driving behaviors that can lead to crash involvement. Many self-report measurement scales are used to assess such risky behaviors. This study is aimed to understand the risky driving behaviors of young adults in Qatar and how such behaviors are associated with crash involvement. This was achieved through the usage of validated self-report measurement scales adopted for the Arabic context. A nationwide cross-sectional and exploratory study was conducted in Qatar from January to April 2021. Due to the Covid-19 pandemic, the survey was conducted online. Therefore, respondents were selected conveniently. Hence, the study adopted a non-probability sampling method in which convenience and snowball sampling were used. A total of 253 completed questionnaires were received, of which 57.3% were female, and 42.7% were male. Approximately 55.8% of these young drivers were involved in traffic accidents after obtaining their driving license. On average, most young drivers do have some risky driving behavior accompanied by a low tendency to violate traffic laws, and their driving style is not significantly controlled by their personality on the road. The older young drivers are more involved in traffic accidents than the younger drivers, i.e., around 1.5 times more likely. Moreover, a young male driver is 3.2 times less likely to be involved in traffic accidents than a female driver. In addition, males are only 0.309 times as likely as females to be involved in an accident and have approximately a 70% lower likelihood of having an accident versus females. The analysis is complemented with the association between young drivers’ demographic background and psychosocial-behavioral parameters (linking risky driving behavior, personality, and obligation effects on crash involvement). Some interventions are required to improve driving behavior, such as driving apps that are able to monitor and provide corrective feedback.

## 1. Introduction

Road accidents have become one of the most common causes of morbidity and mortality, especially in children and young adults aged 5–29 years [1]. According to the World Health Organization (WHO) [1], approximately 1.3 million die each year due to road accidents, resulting in a loss of around 3% of the country’s gross domestic product (GDP). In the United States, the estimated economic loss to society due to road accidents amounted to $242 billion in 2010 [2]. There are several contributors that lead to road accidents, one of which is human error. The following are some of the risk factors identified by the WHO related to human errors: (1) speeding–every 1% increase in the mean speed increases fatal crash risk probability by 4%, (2) seatbelt usage–usage will reduce fatality by 45–50% for front-seat occupants, (3) driving under the influence of psychotropic substances–can increase risk by about five times, and (4) distracted driving–usage of mobile phones can increase the risk of an accident by 4 folds. Many studies have been published supporting the claim by the WHO, indicating that risky driving behaviors are one of the leading causes of road accidents [3,4,5,6,7,8,9].

It was also reported by the WHO [1] that 73% of all road accident deaths occur in young men aged 25 years and below. This is supported by Scott-Parker et al. [10] that driving violations such as speeding, a form of risky behavior, are very common among young drivers. Fu et al. [11] found that young drivers tend to tailgate vehicles in front of them, while Fernandes et al. [12] showed that young drivers tend to overtake vehicles dangerously. Driver inattention among young drivers is another significant contributor to crashes [13]. Inattention can also be caused by distractions such as mobile phone usage, eating, drinking, and too many people in vehicles [14,15,16,17,18,19]. Such risky behaviors are also evident among young drivers in Qatar. In a study conducted by Awadalla et al. [20], it was found that young drivers in Qatar have higher risks of accidents owing to their risky driving behaviors. In addition, it was found that male drivers had a 10 times higher road mortality than the general population. This was supported by a study by Soliman et al. [21], who found that men under the age of 25 in Qatar, with a low level of education and who use a 4-wheel-drive (4WD) had the worst driving errors, violations, and lapses. Besides this, the importance framing the local accident database with an international approach is crucial, as stated by Casao-Sanz et al. [22]. From Qatar’s Traffic Department, the total number of road fatalities in 2016 was 178, while in 2017, the number was 177, and in 2018 it stood at 168. The main reason for accidents, according to the report, was reckless driving, which is 42.4 percent of the total cases. To support this, in a separate study [23], drivers in Qatar have been identified as high risk for road traffic injuries.

Driving is a complex activity that requires (i) cognitive tasks, such as visual and perceptual stimuli and information processing, and (ii) physical tasks: vehicle control in response to a dynamic environment [24]. Several tools have been used to study and assess driving behavior, and actual road assessment is the ideal approach. However, this endeavor can be costly, stressful, and difficult to conduct for several reasons [24]. One of the easiest and fastest exploratory research approaches is the use of self-report instrument tools. To study risky driving behavior, researchers worldwide have used self-report instrument tools such as the Manchester Driver Behavior Questionnaire (DBQ) [25], Driver Attitude Questionnaire (DAQ) [26], Driving Skill Inventory [27], and Behavior of Novice Young Drivers scale (BNYDS) [28]. Self-reporting instrumental tools are preferred by researchers because they are easy to use and cheap [29]. Among these tools, DBQ is the most widely used [30]. This tool was developed to measure two dimensions: (a) *violations* such as speeding, crossing red lights, aggressive stance, and (b) *errors* such as misjudgment [30]. However, DBQ has recently drawn criticism because of its poor predictive ability [31,32] and the omission of other recent factors that contribute to crash accidents [30]. Many crash-contributing factors, such as the usage of mobile phones, fatigue, and time pressure, are not well captured by existing self-report tools. BNYDS was effective when studying novice young drivers [28]; however, this self-report tool is not easily transferrable to another international driving environment because of the uniqueness of the tool, which was developed for Queensland, Australia [30]. Al Reesi et al. [30] argued that there is a need to design self-report tools that are culturally specific, which led to the development of the first Arab self-report tool, which is a modified version of DBQ and BNYDS [30]. The tool was validated and used to study risky driving behaviors among young Omani drivers.

Personality traits, such as aggression, anxiety, and sensation seeking, are also possible predictors of risky driving behavior [33,34,35,36]. A study by Ulleberg et al. and Spano et al. [33,34] showed that young adult drivers showed more difficulty in self-regulation in relation to personality traits. In a separate study by the Society of Risk Analysis [35], it was found that an angry personality leads to aggressive driving behavior, as reported in Roseborough et al. [36]; they developed a model that showed that drivers’ belief in an unjust world leads to driving injustice in the form of aggressive retaliatory driving. Such studies have demonstrated that personality traits are important factors to consider.

In view of the above, an exploratory study was conducted to understand the driving behavior of young adults in Qatar and its relation to traffic accidents by addressing the following hypothesis:

**Hypothesis** **1.**
*Positive personality and positive obligation towards traffic laws have a significant positive impact on safe driving behavior.*


**Hypothesis** **2.**
*Positive personality and positive obligation towards traffic laws have a significant impact on reducing accident involvement.*


**Hypothesis** **3.**
*Each driving behavior factor (DB) has a significant impact on accident involvement.*


**Hypothesis** **4.**
*Demographics traits have significant positive impacts on accident involvement.*


Hence, the objectives of this study are to (i) explore the validity and reliability of the tool by Al Reesi [30] in the Qatari context and (ii) use the validated modified self-report to study the risky driving behavior of young drivers in Qatar and its correlation to traffic accidents by addressing the above hypotheses.

## 2. Methods

### 2.1. Study Design

A nationwide cross-sectional and exploratory study was conducted in Qatar from January to April 2021 to measure risky driving behavior, personality, and obligation to traffic laws while driving among young drivers in Qatar. As stated in the previous section, the aim is to develop connections between drivers’ behavior, obligation towards traffic laws, and personality with traffic accidents. The study was approved by the institutional review board (ethics committee) of Qatar University.

### 2.2. Study Population and Sampling

The total population of Qatar is approximately 2.6 million as of June 2021. The study population was targeted to all young drivers in Qatar from 18 to 25 years, both males and females, which comprise around 300,000 people (11% of the total population). Based on this, the sample size was calculated using the Raosoft^®^ sample size calculator (http://www.raosoft.com/samplesize.html accessed on 30 May 2020). At the 95% confidence level, 5% margin of error, and with a response distribution of 50%, the estimated sample size for this study was 384. A 50% sample was added for the possibility of non-responders, which gave a final sample size of 576. Due to the Covid-19 pandemic, it was impossible to meet the participants physically; as such, the survey was conducted online. We distributed the survey through email blasts, social media, and student leaders. Therefore, respondents were selected conveniently. Hence, the study adopted a non-probability sampling method in which convenience and snowball sampling were used [30,37].

### 2.3. Outcome Measures and Instrument Validation Process

The survey adopted in this study was a validated questionnaire generated by Al Reesi et al. [30], which was a modification of DBQ and BNYDS. The survey was conducted in two languages, English and Arabic. For the Arabic version, a couple of rounds of forward-backward translations were performed to ensure that the original meaning and context of the survey were not lost. The survey had a few parts. The initial part was related to participants’ socio-demographics, such as age, gender, education level, marital status, ethnicity, working status, nationality, religion, and vehicle ownership. The second part of the survey examined the driving history: (a) learning to drive on-road with or without supervision, (b) driving experience in years, (c) frequency of driving a week, and (d) age at which participants started to learn to drive. The third part of the survey consisted of questions pertaining to seven (7) risky driving behaviors: *transient violation*, *mood driving*, *speeding*, *fatigue*, *distracted driving*, *seatbelt usage*, and *tailgating*. The fourth and fifth parts of the survey looked at the personality nature and obligation towards traffic laws while driving. The last section of the survey focused on the crash history of the participants. The researchers utilized the *Qualtrics* survey tool for this study since the entire survey was conducted online. The following is a summary of the breakdown of the survey tool:Part 1: Socio-demographics (8 questions)Part 2: Driving history (5 questions)Part 3: Driving behavior (40 questions)
○*DB1–Transient Violation* (17 questions–related to risky maneuvers such as running over a red light, overtaking from the right-hand side, illegal U-turns, putting other vehicles at risk, etc.)○*DB2–Mood driving* (8 questions–related to getting angry while driving, honking, driving fast when in a bad mood, etc.)○*DB3–Speeding* (4 questions–related to speed limits)○*DB4–Fatigue* (3 questions related to tiredness, taking breaks, etc.)○*DB5–Distracted driving* (4 questions related to mobile usage, eating/drinking while driving)○*DB6–Seatbelt usage* (two questions related to the usage of seatbelts while driving)○*DB7–Tailgating* (two questions related to driving close to vehicles in front of the driver)Part 4: Personality (13 questions)
○This part contains questions regarding the persons’ personality while driving, such as prediction of driving consequences, worry about making mistakes, worry about offending others on the road, and aggressiveness while driving.Part 5: Obligation to follow traffic law (9 questions)
○This part seeks to understand the participant’s obligation to follow traffic laws while driving. Questions such as the need to follow traffic rules, the importance of some rules, etc.
Part 6: Traffic Accidents (3 questions)
○Q1–Have you ever been in car accidents○Q2–How many accidents have you been involved in as a driver since getting your driving license?○Q3–From the accidents that you have been involved in, how many were your faults?

### 2.4. Data Collection Procedure

We distributed the survey through email blasts, social media, and student leaders. We managed to obtain 558 respondents; however, after filtering for completeness in responses, a total of 253 responses were identified. This corresponds to a sampling error of ±7% at a 95% confidence level. A similar sample size was used by Roseborough et al. [36], and Moataz et al. [38].

### 2.5. Pilot Study

Before the survey was launched, a pilot study was conducted to examine the content validity, readability, and clarity of each item in the survey. A total of 20 participants participated in the pilot study. Slight changes were made to the language to enhance clarity and readability based on the comments received from the participants. The survey was distributed to the target population.

### 2.6. Statistical Analysis

The Statistical Package for the Social Sciences (SPSS) version 27 (IBM Corp. Released 2020. IBM SPSS Statistics for Windows version 27.0. Armonk, NY: IBM Corp.) was used to analyze the data. SPSS was used to conduct a descriptive analysis of the data obtained (mean [standard deviation, SD], median [interquartile range (IQR)], frequency [%]). This was followed by factorial validity using Cronbach’s alpha to measure internal consistency (interrelatedness of survey items). An alpha coefficient of 0.7 above is considered acceptable [30]. Normality tests were performed for continuous outcome variables. The Mann-Whitney test, Chi-square test, Spearman-rho test, and multiple regression analysis (i.e., linear and logistic) were used. The analysis continues with the association between young drivers’ demographic background and psychosocial-behavioral parameters (linking risky driving behavior, personality, and obligation effects on crash involvement). The a priori significance level was set at 0.05.

## 3. Results

### 3.1. Background Characteristics

Table 1 shows the respondents’ background profiles. The majority were female (*n* = 145, 57.3%) and 21 years old (*n* = 57, 22.5%). The mean age (with standard deviation, sd) of male and female drivers was 20.59 (2.18) and 21.17 (1.96), respectively, with *p* = 0.032 (less than 0.05). There were fewer male drivers than female drivers and significantly younger male than female drivers. In general, most of the respondents (*n* = 205, 80%) drove their vehicles for at least three days per week. The driving histories of the respondents are listed in Table 2. It has been reported that they started to learn to drive from 10 to 24 years of age, with a majority of them (*n* = 89, 34.9%) at the age of 18, followed by 18.8% (*n* = 48) at the age of 17. Approximately 45% (*n* = 114) admitted that they had driven a car without supervision prior to obtaining a driving license. Other major responses indicated that the car that they were driving belonged to their parents (*n* = 147, 58.1%), they learned how to drive a car in a driving school (*n* = 115, 45.5%), and 141 (55.7%) drove a car daily.

Table 3 shows the experiences of traffic accident involvement among young drivers. It shows that 55.8% of these young drivers were involved in some sort of traffic accident after obtaining their driving license. From the median statistics, every one out of two accidents (50%) is the respondents’ fault. However, according to their driving style, approximately 59.7% of the respondents generally think that they are unlikely to be involved in an accident in the coming year. This can be linked to the phenomena of “overconfidence”, as reported by Wohleber and Matthews [39]. Such an overconfidence attitude makes people underestimate risk, and this may contribute to risky behaviors such as speeding and traffic violations. The survey data clearly support the overconfident nature of young adults and their inverse performance on the road.

### 3.2. Psychometric Properties

The quality of the instrument, that is, the reliability measure, was measured using Cronbach’s alpha and is shown in Table 4. The composite risky behavior, which consists of 40 questions, has an internal consistency of α 0.942. Among the seven factors describing risky behavior (DB 1 to DB 7), seatbelt usage had the highest internal consistency (0.898), followed by transient violations (0.873). The lowest fatigue was observed at α = 0.735. The obligation and personality factors have internal consistencies of α = 0.880 and α = 0.470, respectively. The data for overall risky behavior reports an exposure average of 2.24 per item, with the maximum being around 4.325. This shows that, in general, most drivers are involved in risky behaviors at some level. The same can be said about the obligation of traffic laws while driving, where the average exposure per item for an obligation is 2.33, with a maximum of five per item. These findings are consistent with those of other studies reported by Qatar [20,21]. For personality, the average score per item was 3.38, indicating that to some extent, the personality of the participant while driving played a role.

### 3.3. Predicting Crash Involvement

The relationship of the demographic attributes of the young drivers with the psychosocial-behavioral parameters is shown in Table 5. The significant factors were later used in the multiple regression analyses: linear and binary logistic regression. It was found that DB4 (fatigue) and DB6 (seatbelt) were highly associated with gender. When comparing males and females for DB4, the median score for males was 7.0 (interquartile range 4.0), and for females was 5.0 (interquartile range 4.0). It tends to show that male drivers do not take fatigue seriously. For DB6, the males had a median score of 4.0 compared to females scoring 3.0. This shows that males have low regard for using seat belts, which confirms previous findings [38,40]. In terms of nationality, the Qataris had a higher tendency to not use seat belts compared to the non-Qataris (median score of 4.0 vs. 3.0). However, the non-Qataris had a higher tendency for tailgating (median score of 5.0 vs. 4.0). From the analysis, it was found that drivers with tertiary education have higher risky behavior in terms of transient violation (DB1), mood driving (DB2), and fatigue (DB3) compared to drivers in high school. This can be related to the phenomenon of overconfidence, as explained by Ryan et al. [40]. An additional relationship was tested between questions “As a driver, have you ever been in car accidents?” and “Have you gotten any traffic violations during the last 12 months?” The findings showed a significant association between the two questions (*p* = 0.001); 66.7% (*n* = 80 out of 120) of the respondents who had experienced traffic violations in the past 12 months were in car accidents. In contrast, 55.1% (*n* = 70 out of 127) of the respondents who did not receive any traffic violations did not experience any car accidents.

The results in Table 6 explain the association between different psychosocial-behavioral domains/variables. All associations between the predictors were statistically significant (*p* < 0.01). All factors under the behavioral domains were significantly correlated (*p* < 0.001). The overall behavior (OB) was found to be significantly associated (moderate) with the driver’s personality (*p* = 0.000) and obligation (*p* = 0.000). In addition, the traffic accident factor (Q3) was associated with all other factors (*p* ˃ 0.05). Significant factors were considered in the multiple regression model. Two regression models were established: (i) a multiple linear regression model based on the number of faults when involved in an accident, and (ii) a multiple binary logistic regression model based on the response if the young driver was involved in a car accident. The multiple linear regression model was significant (F = 2.571, *p* = 0.007), and only DB1, DB2, DB3, DB4, DB5, DB7, obligation, and overall behavior were significant predictors (adjusted R^2^ = 0.104, symbolizing weak correlation and lack of fit for the model). The multiple linear regression model is shown in Equation (1). This value shows that these factors contribute approximately 10% to traffic accidents (total variation for the dependent variable that could be explained by the independent variables).
Model: Number of faults when involved in an accident = (−2.741) + 0.073 (transient violations) + 0.061 (mood driving) + 0.190 (speeding) + 0.234 (fatigue) + 0.155 (distracted driving) + 0.313 (close following) + 0.028 (overall behavior) + 0.091 (obligation)(1)

Table 7 and Table 8 below indicate the relationship between behaviors, personality, and obligation with traffic accident involvement during the last 12 months. Gender, age, educational level, transient violations, speeding, fatigue, distracted driving, seatbelt usage, and overall behavior were considered in the model. Only age and sex were significant predictors. The multiple logistic regression model that can be developed from the above analysis is shown in Equation (2):Model: Traffic accident = (−8.564) + 0.389 (Age) + (−1.176) (Gender)(2)

With multiple independent variables in the equation, the coefficient indicates the extent to which the dependent variable is expected to increase/decrease (affected, depending on the coefficient) when the independent variable increases by one, holding all the other independent variables constant. This study measured previous and collective psychosocial –behavioral parameters related to driving that influence the assessment of causality towards traffic accidents and violations. Thus, it has a temporal effect.

## 4. Discussion

From the analysis and with reference to Table 4, it is apparent that the young drivers in Qatar rated between “Hardly Ever” to “Occasionally” for the following risky driving traits: *transient violation, mood driving, speeding, fatigue, distracted driving, seatbelt usage, and tailgating*. Similar can be said for personality and obligation to abide by traffic laws. Based on these findings, this group of respondents is obviously not high-risk takers while on the road. This is because of the sampling approach adopted in this study. From the regression models, it was found that the likelihood of committing traffic violations is associated with driving behavior and attitude to abide by traffic laws. There was no strong association found with personality traits, which contradicts the findings reported by [33,34,35,36].

In a study conducted by Al Reesi [30], he found that nearly 60.7% of those involved in traffic accidents were at fault. These are close to the findings obtained in Qatar, which reported that 50% were at fault. With regard to gender, young Omani males had a higher likelihood of being involved in accidents than females. This is the opposite of what we found in Qatar. Young males in Qatar are only 0.309 times as likely as females to be involved in an accident and approximately 70% less likely to have an accident than females. This result can be influenced by the relatively high percentage of female drivers who responded to the questionnaire. Female drivers are prone to accidents connected to maneuvering and control of traffic situations, whereas males are mostly connected to high-speed accidents, resulting in severe injuries and fatalities [41]. Based on the responses, the male respondents were not really engaged in high-speed driving, indicating that fewer accidents were reported. The older young drivers are more involved in traffic accidents than the younger drivers, i.e., around 1.5 times more likely. Moreover, a young male driver is 3.2 times less likely to be involved in traffic accidents than a female driver. The current study showed a significant association between traffic violations and car accidents. This suggests that risky driving behavior, which causes traffic violations, is closely associated with traffic accidents. Similar findings were found by Al-Reesi [30], who examined young Omani drivers.

The following are the responses to the hypotheses posted at the beginning of this study:

**Hypothesis** **1.**
*Positive personality and positive obligation towards traffic laws have a significant positive impact on safe driving behavior (linked to overall behavior).*


Outcome: The study found that a positive obligation towards traffic laws has a positive impact on safe driving behavior. However, the role of personality was not well understood from the statistical analysis, and it remains inconclusive.

**Hypothesis** **2.**
*Positive personality and positive obligation towards traffic laws have a significant impact on reducing accident involvement.*


Outcome: A positive obligation towards traffic laws has a significant impact on reducing accident involvement. The role of personality traits was not found to be an important influencing factor for risky behaviors and car accidents. This is contrary to the findings of other researchers [42], in which personality traits are indeed significant predictors of risk-taking while driving.

**Hypothesis** **3.**
*Each driving behavior factor (DB) has a significant impact on accident involvement.*


Outcome: Based on the statistical analysis, driving behavior has a significant impact on the number of traffic violations, but not on accidents.

**Hypothesis** **4.**
*Demographics traits have significant positive impacts on accident involvement.*


Outcome: The only demographic traits that had a positive impact on accident involvement were the age of the driver and gender.

### 4.1. Limitations

The study presented here is based on a survey conducted with young adult drivers in Qatar and may not be reflective of the wider driving population. Some disparities may exist between the numbers presented here and the official crash records. The final response rate was lower than the initial estimated sample size. There is also the potential for social desirability bias due to the sampling technique adopted in this study. Another potential disadvantage is a lack of realism. The results of this study are influenced by the relatively high percentage of young female drivers as respondents, considering the usual distribution of drivers based on gender, not only in Arabic countries but in general, are male. Despite the above limitations, this study has to some extent, revealed the risky driving behavior among young adults in Qatar.

### 4.2. Future Work and Recommendations:

The results provide support for possible intervention strategies that can be employed by driver education programs to reduce any sort of aggressive driving. A longitudinal research design would allow for an examination of the actual driving behavior of young adults, and perhaps this can be done by using telematics data.

## 5. Conclusions

The psychometric properties of the tool used in this study were found to be valid and reliable. It was found that 55.8% of these young drivers were involved in some sort of traffic accident after obtaining their driving license; from these, every 1 out of 2 accidents is the fault of the young driver. The older young drivers are more involved, i.e., around 1.5 times more likely, to be involved in traffic accidents than the younger drivers. Moreover, a young male driver is 3.2 times less likely to be involved in traffic accidents than a female driver. In addition, males are only 0.309 times as likely as females to be involved in an accident and approximately 70% less likely to have an accident versus females. From the results obtained here, the self-study tool has demonstrated that young adult drivers in Qatar do have some risky driving behavior issues accompanied by a tendency to not comply with traffic regulations while driving. Some of these young drivers are controlled by their aggressive personalities while on the road. Some interventions are required to improve driving behavior in Qatar, such as driving apps that are able to monitor and provide corrective feedback.

## Figures and Tables

**Table 1 ijerph-19-00514-t001:** Young drivers’ background characteristics (N _total_ = 253).

Item	Description	*N*, (%)
Gender	Male	108 (42.7)
	Female	145 (57.3)
Age	18	43 (17.0)
	19	38 (15.0)
	20	30 (11.9)
	21	57 (22.5)
	22	28 (11.1)
	23	23 (9.1)
	24	16 (6.3)
	25	18 (7.1)
Nationality	Qatari	144 (56.9)
	Non-Qatari	109 (43.1)
Ethnicity	Arab	217 (85.1)
	Non-Arab	36 (14.9)
Education Level	Secondary	90 (35.6)
	Tertiary	163 (64.4)

**Table 2 ijerph-19-00514-t002:** Young drivers’ driving history.

Item	Description	N, (%)
At what age did you learn how to drive?	Below 17	71 (28.9)
	17 and above	175 (71.1)
Before you got your driver’s license, did you ever drive without supervision?	No	139 (54.9)
	Yes	114 (45.1)
Who is the owner of the car you are driving?	Father/Mother	144 (56.9)
	Brother/Sister	109 (43.1)
	Self	217 (85.1)
	Others	36 (14.9)
How did you learn to drive?	In a driving school	115 (45.5)
	In the city streets	32 (12.6)
	In the desert	69 (27.3)
	Self-learned	26 (10.3)
	Others	11 (4.3)
How many days do you drive a car per week?	3 to 6 days a week	64 (25.3)
	Daily	141 (55.7)
	Once a week	15 (5.9)
	Twice a week	33 (13.1)

**Table 3 ijerph-19-00514-t003:** Young drivers’ experience with crash involvement.

Item		N (%)	Mean (SD)	Median (IQR)
Have you ever been in car accidents	Yes	139 (55.8)		
	No	110 (44.2)		
How many accidents have you been involved in as a driver since getting your driving license?		136 (53.8)	2.79 (3.80)	2.00 (1.00–3.00)
From the accidents that you have been involved in, how many were your fault?		137 (54.2)	1.68 (2.61)	1.00 (0.50–2.00)
Thinking generally about your driving style, how likely are you to cause an accident next year?	Very Unlikely	59 (23.3)		
	Unlikely	92 (36.4)		
	Unsure	76 (30.0)		
	Likely	13 (5.1)		
	Very Likely	7 (2.8)		

**Table 4 ijerph-19-00514-t004:** Reliability measures of survey tool.

Factors	N	Cronbach’s Alpha (α)	Score Range	Mean (SD)	Median (IQR)	Min–Max
**Overall Behavior**	40	0.942	40 to 200	89.55 (25.10)	89.0 (69.0–107.0)	40.0–173.0
● DB1 (Transient Violations)	17	0.873	17 to 85	29.90 (9.08)	29.0 (23.0–35.0)	17.0–64.0
● DB 2 (Mood Driving)	8	0.854	8 to 40	21.48 (7.36)	21.0 (16.0–27.0)	8.0–40.0
● DB 3 (Speeding)	4	0.839	4 to 20	10.79 (3.86)	11.0 (8.0–13.0)	4.0–20.0
● DB 4 (Fatigue)	3	0.735	3 to 15	6.79 (2.82)	6.0 (4.0–9.0)	3.0–15.0
● DB 5 (Distracted Driving)	4	0.857	4 to 20	11.52 (4.37)	11.0 (8.0–15.0)	4.0–20.0
● DB 6 (Seatbelt Usage)	2	0.898	2 to 10	4.62 (2.78)	4.0 (2.0–7.0)	2.0–10.0
● DB 7 (Tailgating)	2	0.780	2 to 10	4.45 (1.98)	4.0 (3.0–6.0)	2.0–10.0
**Personality**	13	0.470	13 to 65	44.35 (5.87)	44.0 (40.0–48.0)	25.00–61.0
**Obligation**	9	0.880	9 to 45	20.95 (7.40)	21.0 (15.6–26.0)	9.0–45.0

Note: DB = risky driving behavior, IQR = inter quartile range, SD = standard deviation.

**Table 5 ijerph-19-00514-t005:** Association between young drivers’ demographic background and psychosocial-behavioral parameters.

Item		DB1	DB2	DB3	DB4	DB5	DB6	DB7	Overall
**Gender ***	Male	0.484	0.191	0.338	0.002	0.600	0.029	0.280	0.617
	Female								
**Nationality ***	Qatari	0.997	0.751	0.420	0.868	0.374	0.003	0.043	0.895
	Non-Qatari								
**Education Level ***	Secondary Tertiary	0.008	0.039	0.090	0.011	0.044	0.380	0.226	0.007

Note: Mann-Whitney test * was used at an alpha level of 0.05.

**Table 6 ijerph-19-00514-t006:** Association between psychosocial-behavioral parameters and traffic accidents.

	Transient Violation	Mood Driving	Speeding	Fatigue	Distracted Driving	Seatbelt Usage	Tailgating	Personality	Obligation	Overall (OB)	Q3
Transient violations, DB1	--										
	.										
	253										
Mood driving, DB2	0.665	--									
	0.000	.									
	253	253									
Speeding, DB3	0.592	0.622	--								
	0.000	0.000	.								
	253	253	253								
Fatigue, DB4	0.561	0.492	0.341	--							
	0.000	0.000	0.000	.							
	253	253	253	253							
Distracted driving DB5	0.637	0.710	0.488	0.498	--						
	0.000	0.000	0.000	0.000	.						
	253	253	253	253	253						
Seatbelt usage, DB6	0.429	0.424	0.327	0.242	0.506	--					
	0.000	0.000	0.000	0.000	0.000	.					
	253	253	253	253	253	253					
Close following, DB7	0.602	0.443	0.443	0.358	0.358	0.254	--				
	0.000	0.000	0.000	0.000	0.000	0.000	.				
	253	253	253	253	253	253	253				
Personality	−0.338	−0.187	−0.259	−0.139	−0.163	−0.0183	−0.164	--			
	0.000	0.003	0.000	0.027	0.010	0.003	0.009	.			
	253	253	253	253	253	253	253	253			
Obligation	0.543	0.461	0.453	0.309	0.402	0.216	0.362	−0.323	--		
	0.000	0.000	0.000	0.000	0.000	0.001	0.000	0.000	.		
	242	242	242	242	242	242	242	242	242		
Overall behavior	0.899	0.860	0.709	0.653	0.804	0.551	0.605	−0.306	0.554	--	
	0.000	0.000	0.000	0.000	0.000	0.000	0.000	0.000	0.000	.	
	253	253	253	253	253	253	253	253	242	253	
Fault at accident (Q3)	0.356	0.235	0.262	0.316	0.342	0.062	0.221	−0.012	0.240	0.341	--
	0.000	0.006	0.002	0.000	0.000	0.473	0.010	0.891	0.005	0.000	.
	137	137	137	137	137	137	137	137	137	137	137

**Table 7 ijerph-19-00514-t007:** Causal relationship between psychosocial-behavioral parameters and number of faults when involved in an accident.

Item	Unstandardized Coefficients		Standardized Coefficients			95.0% CI for B	
	*B*	*Std. Error*	*Beta*	*T*	*Sig.*	*Lower Bound*	*Upper Bound*
Constant	−2.741	2.367		−1.158	0.249	−7.424	1.943
Transient violations	0.073	0.023	0.263	3.164	0.002	0.027	0.118
Mood driving	0.061	0.029	0.178	2.102	0.037	0.004	0.119
Speeding	0.190	0.058	0.272	3.280	0.001	0.076	0.305
Fatigue	0.234	0.087	0.225	2.684	0.008	0.061	0.406
Distracted driving	0.155	0.049	0.262	3.159	0.002	0.058	0.252
Seatbelt usage	0.073	0.078	0.081	0.940	0.349	−0.080	0.226
Tailgating	0.313	0.110	0.238	2.849	0.005	0.096	0.530
Personality	0.021	0.026	0.070	0.813	0.418	−0.031	0.074
Obligation	0.091	0.028	0.271	3.265	0.001	0.036	0.146
Overall behavior	0.028	0.008	0.279	3.381	0.001	0.012	0.044

**Table 8 ijerph-19-00514-t008:** Causal relationship between psychosocial-behavioral parameters and number of faults when involved in an accident.

Item	B	Std. Error	Wald	Df	Sig.	OR	95% CI for OR
Constant	−8.564	1.838	21.708	1	0.000	0.000	
Age	0.389	0.090	18.853	1	0.000	1.476	1.238–1.759
Gender code	−1.176	0.330	12.654	1	0.000	0.309	0.162–0.590
Education level code	0.175	0.352	0.246	1	0.620	1.191	0.597–2.376
Transient violations	0.034	0.044	0.599	1	0.439	1.035	0.949−1.128
Speeding	0.136	0.072	3.593	1	0.058	1.145	0.995–1.318
Fatigue	0.017	0.079	0.045	1	0.833	1.017	0.871–1.187
Distracted driving	0.115	0.069	2.765	1	0.096	1.122	0.980–1.284
Seatbelt usage	0.108	0.077	1.972	1	0.160	1.114	0.958–1.296
Overall Behavior	−0.036	0.032	1.286	1	0.257	0.965	0.907–1.026

## Data Availability

The data presented in this study are available on request from the corresponding author. The data are not publicly available due to some restrictions, and they are only available on reasonable request.

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
