# Peer review of "Understanding Traffic Accidents among Young Drivers in Qatar"

_ijerph, 2022, doi:10.3390/ijerph19010514_

Round 1

Reviewer 1 Report

The manuscript has been improved. The authors have replied to every questions arised

Author Response

We thank the reviewer for his positive support towards the manuscript. 

Reviewer 2 Report

Congratulations to the Authors as the study has been significantly improved after resubmission, following the indications of the Reviewers.

I believe that it can now be considered for possible publication in IJERPH.

Best regards,

the Reviewer

Author Response

(The authors gave the same response as above.)

Reviewer 3 Report

The purpose of this paper is to analyze risky driving behaviors in the group of young drivers. The methodology is based on a survey and is suitable for the research study. Taken into account this positive assessment as a whole, I would like to mention some minor aspects (or reflections) that can be appropriate to clarify in order to be published. Some of them will help the reader to better understand the whole research:

Main remarks:

1) Abstract

The abstract does not reflect the content of the paper. The methodology (survey) is not clear. The main aim of the paper is also not clear. Please, review the sentence:  “…This study aimed to understand how risky driving behavior, personality, and obligation for traffic laws are (????) correlated with traffic accidents”. The study does not include a traffic accident database. The analysis is complemented with the association between young drivers’ demographic background and psychosocial-behavioral parameters (linking risky driving behavior, personality, and obligation effects on crash involvement).

2) Introduction

Accidents involving young drivers can be also studied through the statistical analysis of the national accident database. Qatar accident database is not mentioned in the introduction. The methodology does not include an exploratory analysis of the accident data base in Qatar. We do not know how many fatal and seriously injured accidents (involving young driver) take place in Qatar. A minimum information is required for the research. The quality and

EDITOR

As a conclusion, I believe the paper is and could contribute to the literature. After revisions are made to address the main comments, I think the paper may be accepted for publication in the Int. Journal of  Environment  Research and  Public Health.

The purpose of this paper is to analyze risky driving behaviors in the group of young drivers. The methodology is based on a survey and is suitable for the research study. Taken into account this positive assessment as a whole, I would like to mention some minor aspects (or reflections) that can be appropriate to clarify in order to be published. Some of them will help the reader to better understand the whole research:

Main remarks:

1) Abstract

The abstract does not reflect the content of the paper. The methodology (survey) is not clear. The main aim of the paper is also not clear. Please, review the sentence:  “…This study aimed to understand how risky driving behavior, personality, and obligation for traffic laws are (????) correlated with traffic accidents”. The study does not include a traffic accident database. The analysis is complemented with the association between young drivers’ demographic background and psychosocial-behavioral parameters (linking risky driving behavior, personality, and obligation effects on crash involvement).

2) Introduction

Accidents involving young drivers can be also studied through the statistical analysis of the national accident database. Qatar accident database is not mentioned in the introduction. The methodology does not include an exploratory analysis of the accident data base in Qatar. We do not know many fatal and seriously injured accidents (involving young driver) take place in Qatar. A minimum information is required for the research. 

The quality of an accident database determines the research analysis in each country. Please, try to frame the Qatar accident database with an international approach. An International comparison between elements in the official road crash databases can be found in the following reference, that can be used in the paper:

Casado-Sanz, N.; Guirao, B.; Lara Galera, A.; Attard, M. Investigating the Risk Factors Associated with the Severity of the Pedestrians Injured on Spanish Crosstown Roads. Sustainability 201911, 5194. https://doi.org/10.3390/su11195194

nd seriously injured accidents (involving young driver) take place in Qatar.

Author Response

1) Abstract

The abstract does not reflect the content of the paper. The methodology (survey) is not clear. The main aim of the paper is also not clear. Please, review the sentence:  “…This study aimed to understand how risky driving behavior, personality, and obligation for traffic laws are (????) correlated with traffic accidents”. The study does not include a traffic accident database. The analysis is complemented with the association between young drivers’ demographic background and psychosocial-behavioral parameters (linking risky driving behavior, personality, and obligation effects on crash involvement).

Author Response: We thank the reviewer for this excellent suggestion. The Abstract was improved as suggested by the reviewer. This is highlighted green in the manuscript.

2) Introduction

Accidents involving young drivers can be also studied through the statistical analysis of the national accident database. Qatar accident database is not mentioned in the introduction. The methodology does not include an exploratory analysis of the accident data base in Qatar. We do not know how many fatal and seriously injured accidents (involving young driver) take place in Qatar. A minimum information is required for the research. 

Author Response: We thank the reviewer for this excellent suggestion. The Introduction was improved as suggested by the reviewer. This is highlighted green in the manuscript. the suggested reference to the database was also included in the paper.